# Optimization of Ultrasonic-Assisted Extraction of Chlorogenic Acid from Tobacco Waste

**DOI:** 10.3390/ijerph19031555

**Published:** 2022-01-29

**Authors:** Guoming Zeng, Yujie Ran, Xin Huang, Yan Li, Maolan Zhang, Hui Ding, Yonggang Ma, Hongshuo Ma, Libo Jin, Da Sun

**Affiliations:** 1Chongqing Engineering Laboratory of Nano/Micro Biological Medicine Detection Technology, School of Architecture and Engineering, Chongqing University of Science and Technology, Chongqing 401331, China; 2017015@cqust.edu.cn (G.Z.); glad1232022@163.com (Y.R.); 2019443106@cqust.edu.cn (X.H.); abc05062022@163.com (H.D.); hx17358367180@163.com (H.M.); 2School of Pharmacy, Taizhou Polytechnic College, Taizhou 225300, China; 20066954@cqu.edu.cn (Y.L.); zml@cqu.edu.cn (Y.M.); 3Biomedical Collaborative Innovation Center of Zhejiang Province, Institute of Life Sciences, Wenzhou University, Wenzhou 325035, China; 20160121@wzu.edu.cn

**Keywords:** waste treatment, chlorogenic acid, optimal design, preprocessing, ultrasonic-assisted extraction

## Abstract

Using tobacco waste as raw material, the ultrasonic-assisted extraction of chlorogenic acid was optimized by response surface methodology (RSM). After repeated freezing and thawing of tobacco waste twice, the effect of pH value, ethanol volume fraction, temperature and extraction time on the extraction rate of chlorogenic acid was investigated by a single factor experiment. On the basis of this, the factors affecting the yield of chlorogenic acid were further optimized by using RSM. The optimum extraction conditions for chlorogenic acid were set at pH = 4.1, ethanol volume fraction was 49.57% and extraction time was 2.06 h. Under the above conditions, the extraction rate of chlorogenic acid could reach 0.502%, which was higher than traditional extraction and unpretreated ultrasonic extraction. All these results can be used as a reference for the extraction of effective ingredients in tobacco waste.

## 1. Introduction

China, as a major tobacco producer, accounts for about half of the world’s tobacco production. For tobacco, most of it will be used to produce cigarettes, while about a quarter of the leftover material will be discarded and not reasonably utilized [1]. For this part of tobacco waste, if discarded casually, it may destroy the structure of the soil on the one hand; on the other hand, the harmful components in the tobacco waste will also infiltrate into the groundwater, thus further harming human life [2]. In view of this, if these tobacco wastes can be reasonably used as resources, not only can the above problems be solved, but also a large number of natural chemical raw materials can be obtained, and the income of tobacco farmers can also be increased. In addition to this, it is also in line with China’s policy on the resourcefulness of solid waste.

Numerous studies have demonstrated that tobacco leaves contain a number of active substances, such as saccharides, organic acids, alkaloids and proteins [3,4,5,6,7]. However, there are few studies on the contents of these substances in waste materials, such as tobacco stems [8]. Among many bioactive substances, chlorogenic acid has been widely used in the fields of medicine, food and cosmetics because of its good antibacterial, anti-inflammatory, free radical scavenging and anti-cancer properties [9,10,11]. At present, the extraction of chlorogenic acid in the literature was mainly derived from Chinese herbal medicines, such as honeysuckle, *Eucommia ulmoides* leaves and so on [12,13,14]. Besides, most of them were extracted directly without any preprocessing. The extraction methods of chlorogenic acid mainly involve water extraction [12], alcohol extraction [14], ultrasonic extraction [15] and microwave extraction [16]. Among these methods, ultrasonic extraction of chlorogenic acid has the advantages of a fast mass transfer process and high extraction efficiency, which has attracted the attention of many researchers [17].

In order to further improve the resource utilization of tobacco waste and provide a reference for the extraction of chlorogenic acid from tobacco stems and other wastes. In the present study, chlorogenic acid was extracted from tobacco stems using double freeze-thaw method ultrasonic technology for the first time. The tobacco waste was first repeated freezing and thawing, and then the effect of ultrasonic time, temperature and ethanol content on the yield of chlorogenic acid was investigated using a single factor method. Finally, the response surface methodology (RSM) was used to find the optimal extraction process parameters.

## 2. Materials and Methods

### 2.1. Materials

The tobacco waste samples mainly consisting of tobacco stems were collected from a local cigarette factory in Chongqing, China, in September 2020. Chlorogenic acid standard was purchased from Aladdin Bio-Chem Technology, China. Anhydrous ethanol was purchased from the Chongqing Drug Stock Limited Company (Chongqing, China); the experimental water was double deionized water, and the resistance was 16 MΩ.

### 2.2. Experimental Methods

#### 2.2.1. Standard Curve of Chlorogenic Acid

Accurately weighed 3.5 mg of chlorogenic acid standard in 25 mL brown volumetric flask, and then added 50% ethanol to the mark to prepare the standard solution. After that, chlorogenic acid solutions of 0.5 mL, 1.0 mL, 1.5 mL, 2.0 mL, 2.5 mL and 3.0 mL were transferred into 25 mL brown volumetric bottles and then the chlorogenic acid solutions of different concentrations were obtained by making volume with 50% ethanol. A 50% ethanol solution was also used as the blank control. The absorbance of chlorogenic acid solution with different concentrations was determined by an uv-vis spectrophotometer at 329 nm. The standard curve of Y = 0.044X–0.013, R^2^ = 0.9993 was drawn with the concentration of chlorogenic acid standard as the X-coordinate and the absorbance as the Y-coordinate. This result indicated a linear relationship between the absorbance values and the concentration of chlorogenic acid in the 2.8 ug/mL to 16.8 ug/mL range.

#### 2.2.2. Pretreatment of Tobacco Waste

Tobacco waste was first cleaned under running water and rinsed in deionized water after removing impurities. After that, it was placed in a 70–75 °C electric constant temperature blast drying oven for drying. The dried tobacco waste was crushed and sifted through a 40-mesh sieve, and then collected in a brown jar for further use. Tobacco waste powder (0.5 g) was weighed and then transferred into a 50 mL centrifuge tube. A certain volume of deionized water was added to make it completely cover the tobacco waste powder, sealed and then the chlorogenic acid was extracted by ultrasonic after two cycles of freezing. The first reaction was performed at 40 °C for 30 min, and then freeze-thawed at −20 °C; the second reaction was performed at 40 °C for 30 min, freeze-thawed at −20 °C, and then cooled at room temperature for use.

#### 2.2.3. Extraction of Chlorogenic Acid from Tobacco Waste

The waste tobacco stem solution was dried after the above pretreatment, and then the anhydrous ethanol solution was added in accordance with the preset proportion to ultrasonic extract, centrifugation and collected the supernatant (the specific preparation process is depicted in Figure 1). The obtained supernatant was then diluted 20 times and the absorbance value at 329 nm was detected by UV spectrophotometer. The concentration of chlorogenic acid was calculated according to the standard curve and then substituted into the following equation to obtain the yield of chlorogenic acid.
$$\mathrm{The}\;\mathrm{yield}\;\mathrm{of}\;\mathrm{Chlorogenic}\;\mathrm{acid}=\frac{C_{\left(\mathrm{Chlorogenic}\;\mathrm{acid}\right)}\times \mathrm{Dilution}\;\mathrm{ratio}\times V_{(\mathrm{Extraction}\;\mathrm{Solution})}}{W_{\left(\mathrm{Raw}\;\mathrm{material}\right)}\;}\times 100\%$$

#### 2.2.4. HPLC Analysis

Sample preparation and analysis: cut tobacco leaves, dry at 40 °C, pass through a sieve with a 1.0 mm aperture after crushing, and place them in a sample bottle for later use; accurately weigh 40.0 mg of tobacco powder and place it in two 10 mL plugs for quantitative determination. In the test tube, add 45% ethanol solution, ultrasonically extract at a certain temperature, centrifuge, dilute to 10 mL, filter with a 0.22 um filter, and inject and analyze according to the selected chromatographic conditions. Chromatographic conditions: Column: Econosphere C18 column (5 um, 4.6 mm × 250 mm), column temperature: 30, flow rate: 1 mL/min, mobile phase: methanol, water, acetic acid, gradient elution program: 0- Within 20 min, the volume ratio of methanol: water: acetic acid changed from 96:0:4 to 76:20:4; 20–30 min changed to 0:100:0. Detection wavelength: 340 nm, injection volume: 10 uL. Preparation of standard stock solution: Accurately weigh 10.0 mg of chlorogenic acid standard sample into a 10 mL volumetric flask, dissolve it with 45% ethanol solution to volume, use it as a stock solution, and store it at low temperature.

#### 2.2.5. Analytical Method

The results were collated, counted and plotted using Origin 8.0. The data are expressed as the means ± SD (*n* = 3).

## 3. Results and Discussion

### 3.1. HPLC Experiment

Figure 2 is the standard curve of chlorogenic acid.Draw standard stock solution (5, 2, 0.5, and 0.3 mL) and dilute to 10 mL in turn, with concentrations of 5.2, 2.3, 0.8, and 0.04 mg/10 mL, respectively. Dilute the standard stock solution and the above-mentioned 4 concentrations after dilution. The gradient was injected and analyzed according to the selected chromatographic conditions. In the range of 0.04−10.0 mg/10 mL, linear regression was performed with the peak area ratio (Y) and the mass concentration ratio (X), and the linear regression equation Y = 178.15X−10. 12, correlation coefficient: = 0. 9997, based on the signal-to-noise ratio of 3, the minimum detection limit of chlorogenic acid is 0. 41 × 10^−2^ mg/10 mL.

### 3.2. Single Factor Experiment

After reviewing the literature on the extraction of chlorogenic acid from biomass by ultrasonic method, four aspects, namely the volume fraction of ethanol, ultrasonic extraction time, temperature and pH value of the extraction solution, were chosen to investigate the effect on the extraction rate of chlorogenic acid from tobacco waste in this experiment. Each group was repeated three times and the average values were taken for the graphs to obtain the optimum ultrasonic extraction conditions for chlorogenic acid in tobacco waste.

#### 3.2.1. Effect of Ethanol Volume Fraction on the Yield of Chlorogenic Acid in Tobacco Waste

The effect of ethanol volume fractions of 40%, 50%, 60% and 70% on the extraction yield of chlorogenic acid was examined. The chlorogenic acid yield first increased and then decreased as the volume fraction of ethanol gradually increased, with a maximum at 50% of the volume fraction of ethanol in Figure 3. This was in accordance with the literature reported [18]. This may be because the molecular structure of chlorogenic acid contained a large number of hydroxyl groups, so an appropriate volume fraction of ethanol could promote the dissolution of chlorogenic acid in tobacco waste. However, when the volume fraction of ethanol was further increased, other fat-soluble substances may also be dissolved, thus interfering with the yield of chlorogenic acid. This may also be caused by the decrease in the diffusion capacity of solute as the volume fraction of ethanol increases [19].

#### 3.2.2. Influence of Ultrasonic Temperature on the Yield of Chlorogenic Acid in Tobacco Waste

The effect of ultrasonic extraction temperature of 50 °C, 60 °C, 70 °C and 80 °C on the extraction yield of chlorogenic acid was investigated, and the results are shown in Figure 4. With the ultrasonic extraction temperature increasing, the extraction yield for chlorogenic acid showed an evident rise first and then decreased, achieving a maximum at 70 °C. This result was slightly higher than the 57 °C reported in the literature [20], which may be caused by the rigorous structure of the tobacco stem compared with the tobacco leaf. The molecular motion speed and dissolution rate of chlorogenic acid will increase with the rising temperature. However, when the ultrasonic temperature was too high, it would cause the destruction of chlorogenic acid, resulting in the reduction of the extraction rate.

#### 3.2.3. Effect of pH on the Yield of Chlorogenic Acid in Tobacco Waste

The change in chlorogenic acid yield was studied when the pH value of the extract gradually rose from 3 to 6, and the results are shown in Figure 5. The yield of chlorogenic acid in tobacco waste was the highest when the pH of the extract was 4. This may be because the molecular structure of chlorogenic acid contained both carboxyl and hydroxyl groups, which would decompose as the pH of the extract gradually increased, leading to a decline in the extraction rate.

#### 3.2.4. Influence of Ultrasonic Time on the Yield of Chlorogenic Acid in Tobacco Waste

The ultrasonication extraction times were set at 1.5 h, 2.0 h, 2.5 h and 3.0 h, and the results are shown in Figure 6. The yield of chlorogenic acid increased first and then decreased as ultrasound extraction time went on. When the extraction time increased to 2 h, the yield of chlorogenic acid was the highest. The possible reason was that, with the extension of ultrasonic time, chlorogenic acid was degraded by the thermal effect or photolysis. It was also possible that the prolonged extraction time would lead to the dissolution of other impurities in the tobacco waste, resulting in a decrease in the dissolution rate of chlorogenic acid. In addition, these impurities may also interfere with the experimental measurement.

### 3.3. Experimental Methods

#### 3.3.1. Box—Behnken Design

Although the single factor experiment can optimize the experimental conditions to a certain extent, it could not accurately judge the optimal experimental point. Thus, RSM was chosen to obtain the best process parameters in this study. The design was based on the experimental results of the above single-factor experiments, considering the controllability of experimental operation and using chlorogenic acid yield as the index of investigation. The design results are shown in Table 1 by using the Box-Behnken central combination of Design Expert 8.0.6 software. RSM is an optimization method that can intuitively represent the relationship between experimental conditions and dependent variables by using graphical techniques, so it has been applied in many fields [21,22,23,24].

#### 3.3.2. Analysis of Variance

Quadratic polynomial regression fitting was performed on Box-Behnken’s experimental design and experimental results, and the binary regression equation model was obtained as follows:Chlorogenic acid yield % = −2.88 + 0.37A + 0.07B + 0.57C + 5 × 10^−4^AB + 0.03AC + 1.25 × 10^−4^BC − 0.13A^2^ − 7.59 × 10^−4^B^2^ − 0.08C^2^.

Significance test was performed on the coefficient of the regression model, and the results were shown in Table 2. It can be seen from the table that the model *P* < 0.05, the difference was significant, indicating that the regression model obtained in this experiment had a good fitting degree for the yield of chlorogenic acid in tobacco waste. The R^2^ was 0.9068, indicating that the experimental error was small. According to the value of F, it can be judged that the order of influence of various factors on chlorogenic acid extraction in tobacco waste was as follows: pH of the extraction solution > volume fraction of ethanol > ultrasonic time. The quadratic terms B^2^ and C^2^ were significant, indicating that they have a great influence on the yield of chlorogenic acid. The missing term *p* = 0.0749 > 0.05, showing no significance, indicates that factors other than the model study had little influence on the response value. Therefore, the regression equation obtained in this study can be used to determine the best ultrasonic-assisted extraction process of chlorogenic acid in tobacco waste.

#### 3.3.3. Response Surface Map Analysis

The data in Table 2. were fitted by quadratic multiple regression to obtain the corresponding response surface stereogram, as shown in Figure 7.

The contour line indicated that the chlorogenic acid yield in tobacco waste was the same in the same oval area and gradually decreased from the center to the edge [25,26]. Therefore, as shown in Figure 7, the vertex position of the response surface map indicated the highest extraction yield of chlorogenic acid. The opening of the response surface was downward, indicating that the chlorogenic acid yield increased with the increase of the value of each factor in a certain range. Then, after the maximum respective face value was reached, the chlorogenic acid yield decreased as each factor increased further. According to the p value of the significance test of each variable in Table 2, it can be seen that the influence of each factor on the yield of chlorogenic acid is: C^2^ > B^2^ > A^2^ > C > AC > B > A > AB > BC, indicating that there is a certain relationship between the factor interactions. The optimal extraction conditions of chlorogenic acid from tobacco waste by RSM were as follows: ultrasonic time of 2.06 h, ethanol volume fraction of 49.57%, pH of 4.1, and the theoretical yield of chlorogenic acid was 0.502%. In order to verify the feasibility of the above method and consider the feasibility of actual operation, the conditions were modified and carried out under the following conditions: ultrasonic extraction time of 2.0 h, ethanol volume fraction of 50% and pH value of 4.1. Finally, the yield of chlorogenic acid extracted from tobacco waste was 0.497% under the above conditions, which was very close to the predicted results.

## 4. Conclusions

The tobacco waste was first pre-treated by repeated freeze-thawing before ultrasonic extraction of chlorogenic acid. Then, single-factor experiments and RSM were chosen to optimize the extraction process parameters of chlorogenic acid from tobacco waste. The obtained optimal process conditions were as follows: 2.06 h of ultrasonic extraction time, 49.57% of ethanol by volume, and extracting solution pH of 4.1. Under these conditions, the yield of chlorogenic acid could reach 0.502%. This result indicated that the resource utilization of tobacco waste, such as tobacco stems, was an effective way to develop its added value. In addition, the repeated freeze-thawing method adopted in this study has the advantages of being simple, low energy consumption and low cost when compared with other pretreatment methods, such as ionic liquid and steam blasting, so it could provide a reference for the extraction of chlorogenic acid and even other active ingredients in biomass cells.

## Figures and Tables

**Figure 1 ijerph-19-01555-f001:**
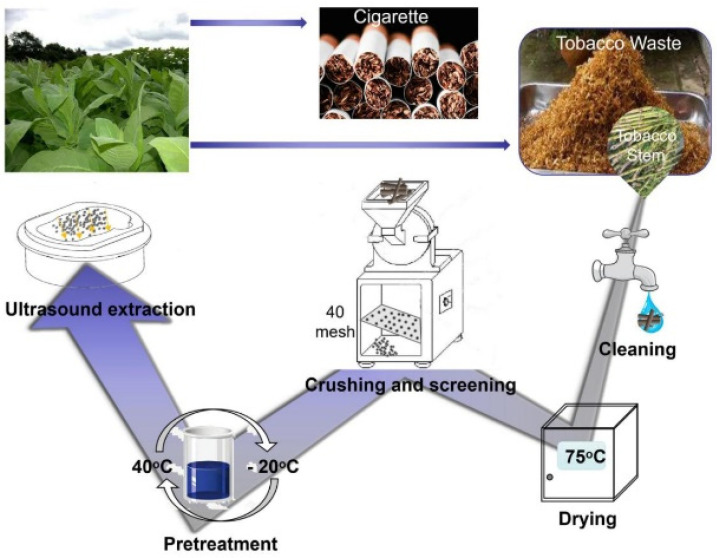
The flow chart of chlorogenic acid extraction from tobacco stem.

**Figure 2 ijerph-19-01555-f002:**
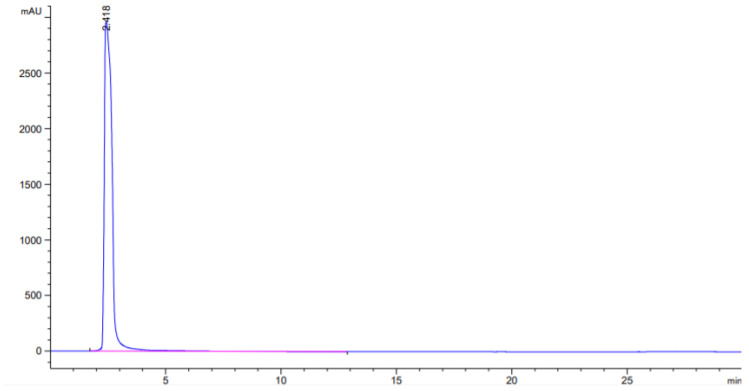
HPLC chromatogram of chlorogenic acid in tobacco.

**Figure 3 ijerph-19-01555-f003:**
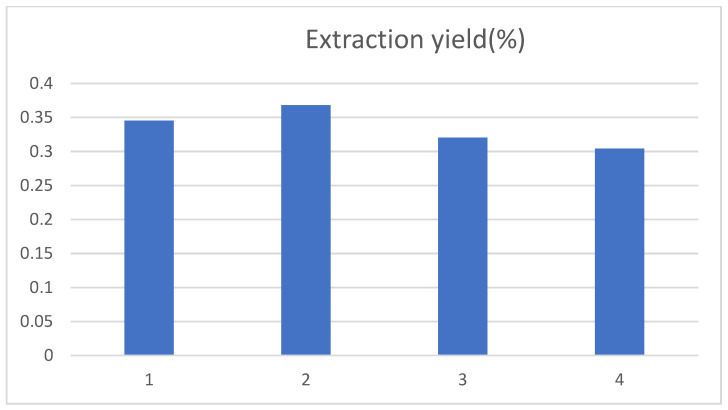
Influence of ethanol volume fraction on the extraction yield of chlorogenic acid, and 1, 2, 3, and 4 represent the ethanol volume fraction of 40%, 50%, 60% and 70%, respectively.

**Figure 4 ijerph-19-01555-f004:**
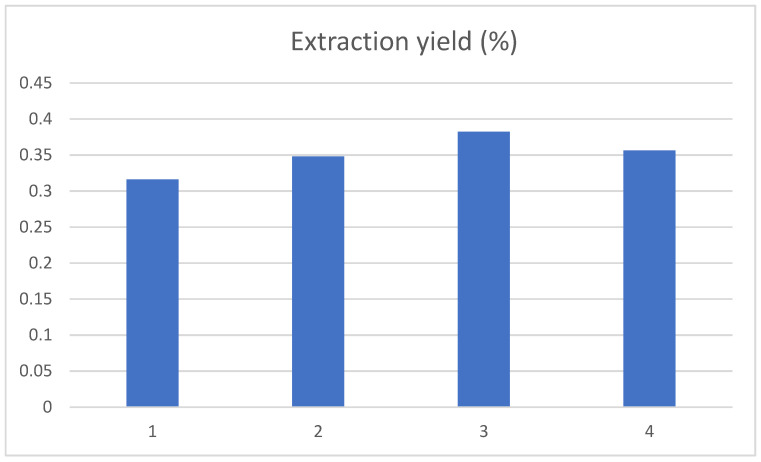
Influence of ultrasonic temperature on the extraction yield of chlorogenic acid, and the 1, 2, 3, and 4 represent the ultrasonic extraction temperature of 50 °C, 60 °C, 70 °C and 80 °C, respectively.

**Figure 5 ijerph-19-01555-f005:**
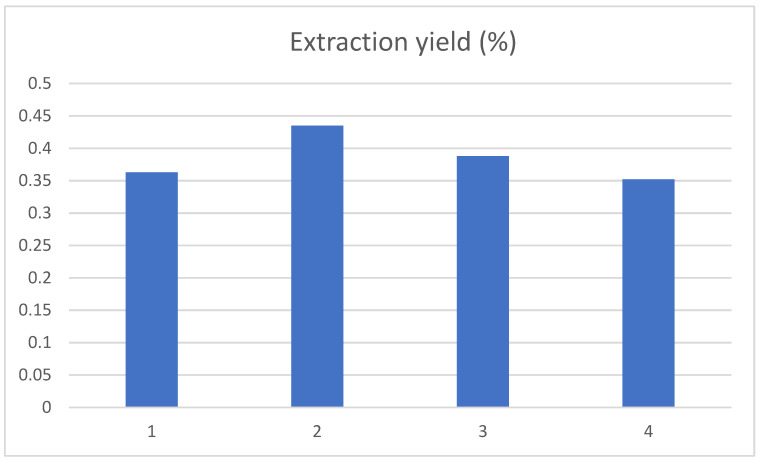
Influence of pH value on the extraction yield of chlorogenic acid, and the 1, 2, 3, and 4 represent the pH value of 3, 4, 5 and 6, respectively.

**Figure 6 ijerph-19-01555-f006:**
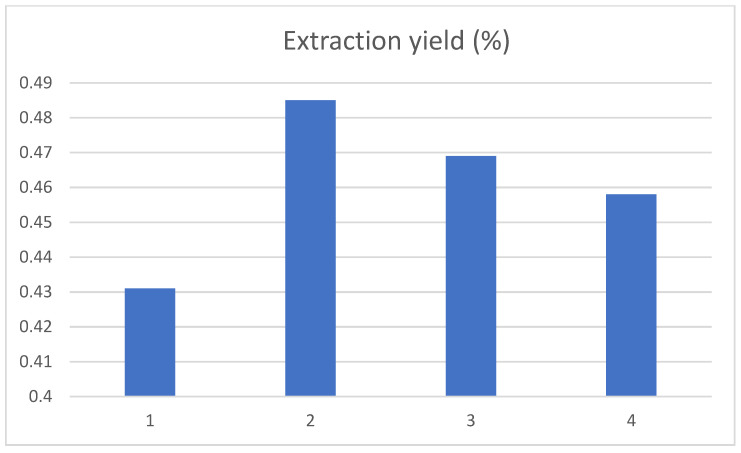
Influence of ultrasound time on the extraction yield of chlorogenic acid, and the 1, 2, 3, and 4 represent the extraction times of 1.5 h, 2.0 h, 2.5 h and 3.0 h, respectively.

**Figure 7 ijerph-19-01555-f007:**
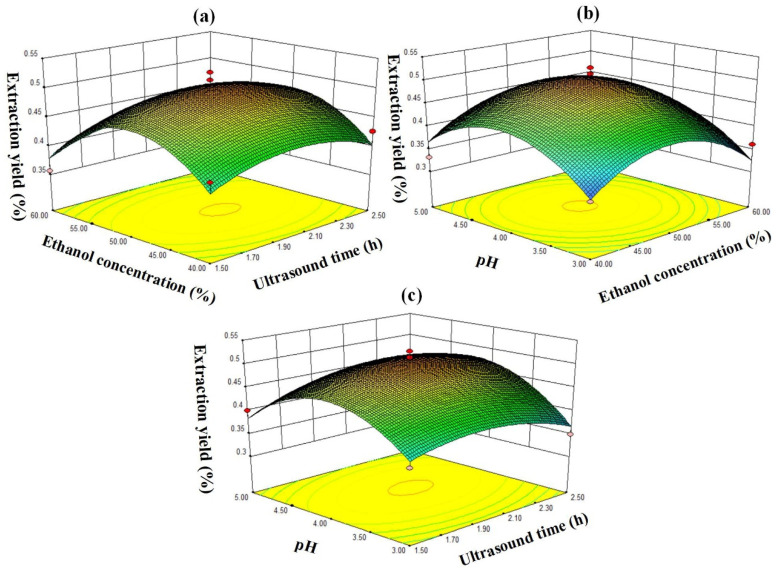
Response surface plots for the effects of (**a**) ethanol concentration vs. extraction time, (**b**) ethanol concentration vs. pH, and (**c**) pH vs. extraction time on the extraction yield of chlorogenic acid.

**Table 1 ijerph-19-01555-t001:** Response surface analysis and results.

Run	Factors			The Yield of Chlorogenic Acid (%)
	A: Ultrasound Time (h)	B: Ethanol Concentration (%)	C: pH Value of Extraction Solution	
1	2.0	40	3.0	0.336
2	2.0	50	4.0	0.486
3	1.5	60	4.0	0.356
4	2.0	50	4.0	0.527
5	2.0	50	4.0	0.492
6	2.5	50	5.0	0.439
7	2.5	50	3.0	0.349
8	1.5	50	3.0	0.374
9	2.5	60	4.0	0.377
10	2.0	60	3.0	0.360
11	2.0	40	5.0	0.331
12	2.0	50	4.0	0.487
13	2.5	40	4.0	0.426
14	1.5	40	4.0	0.415
15	1.5	50	5.0	0.401
16	2.0	60	5.0	0.360
17	2.0	50	4.0	0.514

**Table 2 ijerph-19-01555-t002:** The ANOVA for response surface quadratic model to extraction of chlorogenic acid.

Source	Sum of Squares	DF	Mean Square	F Value	*p* ValueProb > F	Remarks
Model	0.063	9	7.019 × 10^−3^	7.57	0.0071	SignificantNot significant
A-Time	2.531 × 10^−4^	1	2.531 × 10^−4^	0.27	0.6175
B-Concentration	3.781 × 10^−4^	1	3.781 × 10^−4^	0.41	0.5435
C-pH	1.568 × 10^−3^	1	1.568 × 10^−3^	1.69	0.2347
AB	2.500 × 10^−5^	1	2.500 × 10^−5^	0.027	0.8742
AC	9.923 × 10^−4^	1	9.923 × 10^−4^	1.07	0.3354
BC	6.250 × 10^−6^	1	6.250 × 10^−6^	6.738 × 10^−3^	0.9369
A^2^	4.271 × 10^−3^	1	4.271 × 10^−3^	4.61	0.0690
B^2^	0.024	1	0.024	26.12	0.0014
C^2^	0.026	1	0.026	28.05	0.0011
Residual	6.493 × 10^−3^	7	9.275 × 10^−4^		
Lack of Fit	5.146 × 10^−3^	3	1.715 × 10^−3^	5.09	0.0749
Pure Error	1.347 × 10^−3^	4	3.367 × 10^−4^		
Cor Total	0.070	16

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
