# Peer review of "Optimization of Ultrasonic-Assisted Extraction of Chlorogenic Acid from Tobacco Waste"

_ijerph, 2022, doi:10.3390/ijerph19031555_

Round 1
Reviewer 1 Report
Please see in the attached file

Author Response
Dear reviewer,
Thank you for your good comments and suggestions. We have modified the paper ,please check it.
Wish you everything is ok.
Guoming Zeng

Reviewer 2 Report
The article entitled “Optimization of Ultrasonic-Assisted Extraction of Chlorogenic Acid from Tobacco Waste" presents the results of a chlorogenic acid extraction from tobacco stem by ultrasonic technology.
Chlorogenic acid is a very valuable organic compound. It may slow down the secretion of glucose into the bloodstream after a meal. It also increases the sensitivity of cells to insulin. It is a powerful antioxidant. The literature emphasizes the potential antibacterial, antiviral and antifungal properties of this compound. Due to the above-mentioned features, it is sold in some countries as a dietary supplement in an isolated form. Considering the above, the research undertaken is very valuable and fits in with the current trends in the circular economy and the recovery of bioproducts from waste. Unfortunately, the authors unskillfully prepared the MS.
General note:
The introduction is presented correctly, in accordance with the subject however, it requires some minor additions. Numerous scientific articles, in concordance to the topic of the study, were consulted.
Materials and methods section needs improvement. The authors have to describe the methodology in more detail. In the current version, part of the methodology description is given in the description of the results. It needs to be sorted out.
What is its main aim? Moreover, the novelty statement should be emphasized. Which gaps of the existing literature do the study aspires to fill?
Results and discussion does not include discussions. It is rather a confusion between the methodology and the description of the results.
There are some major changes I am suggesting in detailed comments below.
Detailed comments:
“2.2.1. Standard curve of chlorogenic acid” - use italics
“40 ℃ (30min) - -20 ℃ (icing) - 40 ℃ (30min) - -20 ℃ (icing) - room temperature” – wrong format.
“Based on the literature, the ultrasonic power of 500W, solid-liquid ratio of 1:15 (g/mL), extraction pH of 4, ultrasonic temperature of 60℃ and ultrasonic time of 40 min were first selected in this study. According to the method described in 2.2.2 and 2.2.3, the effect of ethanol volume fraction of 40%, 50%, 60% and 70% on the extraction yield of chlorogenic acid was examined and the results were shown in Figure 2.” - this section describes the methodology, not the results. She should be moved.
Figures 2 and 3 need to be changed. In this form, the results are hardly readable.
Why does the "Experimental Methods" subsection suddenly appear in the Results and Discussion section - it should be in the methodology.
The methodology does not describe the statistical methods used, and ANOVA suddenly appears in the Results section. The article requires a rethinking of the layout typical of scientific papers. In the current version, everything is confused and incomprehensible.
Information on page 11 must be deleted!
Author Response
Dear reviewer,
Thank you for your good comments and suggestions. We have modified the paper,please check it.
Wish you everything is ok.
Ware regards
Guoming Zeng

Round 2
Reviewer 2 Report
I appreciate the author efforts on this manuscript, which indeed improve the quality of this manuscript. Particularly, the authors added missing information. Thus, I satisfy the authors' respondence and the revision.